# The Role of *NFAT5* in Immune Response and Antioxidant Defense in the Thick-Shelled Mussel (*Mytilus coruscus*)

**DOI:** 10.3390/ani15050726

**Published:** 2025-03-04

**Authors:** Yijiang Bei, Xirui Si, Wenjun Ma, Pengzhi Qi, Yingying Ye

**Affiliations:** 1Zhejiang Fisheries Technical Extension Center, Hangzhou No. 181, Jingchang Road, Wuchang Street, Yuhang District, Hangzhou 310012, China; zsbij@163.com (Y.B.); mwj8890@163.com (W.M.); 2National Engineering Research Center for Marine Aquaculture, Zhejiang Ocean University, Zhoushan 316022, China; sxr20000715@163.com

**Keywords:** NFAT, Vibrio, immune response, *Mytilus coruscus*

## Abstract

This study reveals, for the first time, the immune function of the *NFAT5* gene (*McNFAT5*) in the thick-shelled mussel (*Mytilus coruscus*). The results show that *McNFAT5* plays a critical role in the mussel’s defense against *Vibrio alginolyticus* infection with its highest expression detected in hemolymphs. The silencing *McNFAT5* significantly reduced antioxidant defense capabilities, including the activities of superoxide dismutase (SOD) and Na^+^/K^+^-ATPase. This research provides valuable insights into the immune regulation and evolutionary mechanisms of bivalves.

## 1. Introduction

Bivalve mollusks, such as mussels, clams, and scallops, are integral components of marine ecosystems [1]. These organisms play a crucial role not only in maintaining marine biodiversity and ecological balance but also hold significant importance in global aquaculture. Due to their high nutritional value and substantial economic potential, bivalve mollusks have become indispensable in the human food chain. However, with environmental changes and anthropogenic impacts, these mollusks are increasingly facing challenges from various pathogenic infections [2], which threaten their survival and reproduction. The invasion of pathogens not only causes significant economic losses in the aquaculture industry but also endangers the stability of marine ecosystems. Therefore, in-depth studies of the immune mechanisms in bivalve mollusks are of paramount scientific and practical value for the conservation and sustainable utilization of these precious resources.

Nuclear Factor of Activated T-cells (NFAT) is an inducible DNA-binding factor that associates with interleukin-2 (IL-2) [3]. The NFAT transcription factor family consists of five members: *NFAT1* (*NFATc2*), *NFAT2* (*NFATc1*), *NFAT3* (*NFATc4*), *NFAT4* (*NFATc3*), and *NFAT5* (TonE-BP or NFATL1) [4], all of which share a highly conserved Rel homology domain (RHD). These factors regulate T-cell tolerance, cancer, adaptive immunity, and innate immunity [5,6,7]. Among these, *NFAT5* stands out as the only member of the NFAT family that is not regulated by calcium ions, which distinguishes it from the others [8]. This unique characteristic is associated with *NFAT5*’s evolutionary divergence, making it the most evolutionarily conserved member of the NFAT family [9]. *NFAT5* is found in both invertebrates and vertebrates [10], and this conservation underscores its specialized function in mediating cellular responses to osmotic stress and other environmental stimuli. *NFAT5*’s evolutionary lineage traces back to the RHD-containing transcription factor family, with its first known occurrence in the sea anemone Nematostella vectensis, suggesting its ancient origins [11].

Early studies primarily focused on the role of *NFAT5* in mammalian cells, particularly in T-cell activation and cellular responses to osmotic stress. Through the use of *NFAT5* knockout mouse models, research demonstrated that *NFAT5* is essential for the survival of T cells in high-sodium environments. In the absence of *NFAT5*, T cells exhibited significant apoptosis and were unable to maintain normal numbers under high-sodium conditions. This suggests that *NFAT5* is a key regulatory factor in maintaining T-cell homeostasis and proliferation, especially under pathological hypernatremia. The absence of *NFAT5* leads to increased T-cell apoptosis and impaired proliferation [12]. Beyond T cells, *NFAT5* also plays a notable role in macrophages. When Toll-like receptors (TLRs) are activated, *NFAT5* is significantly upregulated, particularly in response to pathogens such as LPS (TLR4 activation), indicating that *NFAT5* is a critical regulator of pro-inflammatory gene expression [13]. Additionally, *NFAT5* has been found to regulate the production of cytokines during inflammatory responses. Under hyperosmotic conditions, *NFAT5* functions through the Fyn and p38 signaling pathways, not only aiding in cellular adaptation to osmotic pressure but also helping orchestrate the immune system’s response to infection and inflammation by modulating the expression of pro-inflammatory cytokines such as TNF-α and *IL-6* in inflammatory environments [14]. This multifaceted role of *NFAT5*, particularly its involvement in both osmotic stress responses and immune regulation, highlights its broader significance within the immune system. Its functions extend beyond a simple response to environmental stressors, encompassing vital regulatory mechanisms in immune cell survival, activation, and the inflammatory response. As research continues, the importance of *NFAT5* in immune homeostasis and its potential as a therapeutic target in immune-related disorders, such as autoimmune diseases and chronic inflammation, becomes increasingly apparent.

In recent years, the role of *NFAT5* in immune defense has garnered increasing attention, particularly in invertebrates. For example, in *Branchiostoma belcheri*, an NFAT gene involved in innate immunity was identified, marking the first experimental evidence of an NFAT family member participating in immune responses in an invertebrate [15]. This discovery highlights the evolutionary significance of *NFAT5* in mediating immune defense across a broad range of species. Research on NFAT in invertebrates remains relatively sparse compared to the extensive studies conducted on vertebrates. Unlike vertebrates, invertebrates, including bivalves, lack T cell homologs, which suggest that *NFAT5* may have a macrophage-regulation-oriented role in these organisms. Hemolymphs, the primary immune cells in bivalves, play a pivotal role in the innate immune response. This highlights the need to explore *NFAT5*’s function in macrophage-like cells, such as hemolymphs, which act as key mediators in pathogen recognition and clearance. Additionally, the molecular mechanisms governing immune regulation in mollusks remain largely elusive. Most molluscan cytokines are poorly characterized, and homologs of vertebrate interleukins are largely undetectable in these species. This represents a critical limitation in understanding how *NFAT5* and other transcription factors function within the unique context of invertebrate immunity. Filling these knowledge gaps could provide valuable insights into the evolution of immune defense mechanisms in marine invertebrates. To address this gap, the present study represents the first attempt to clone and characterize a homolog of *NFAT5* (*McNFAT5*) from the thick-shelled mussel (*Mytilus coruscus*), which is a species of significant commercial importance in China’s aquaculture industry. The genomic structure of the *McNFAT5* gene was analyzed, and its role in the innate immune response of *M. coruscus* was investigated, specifically in response to Vibrio infections. The results of this research demonstrated that *McNFAT5* plays a significant role in the innate immune system of *M. coruscus* by responding to Vibrio stimulation, thus highlighting its involvement in the defense against bacterial pathogens. This discovery not only sheds light on the crucial regulatory function of *McNFAT5* in mussel immune responses but also provides valuable new insights into the broader immune mechanisms of marine mollusks.

## 2. Materials and Methods

### 2.1. Animals

Two hundred mussels were collected from the Shengsi Islands, Zhoushan, Zhejiang Province. After thorough cleaning, they were transferred to tanks containing artificial seawater and maintained at 25 ± 1 °C with a salinity of 25‰ (representing the total dissolved salts, not just NaCl) for a one-week acclimation period. During acclimation, they were fed spirulina powder daily, and the seawater was regularly refreshed. Mortality stayed under 5% during this period. To standardize conditions, feeding stopped a day before the experiment, and only healthy mussels were selected for further analysis.

### 2.2. Full-Length Cloning and Bioinformatics Analysis of McNFAT5

We obtained the full-length transcriptome of mussels using Direct RNA Sequencing (DRS) technology based on the Oxford Nanopore platform. From this dataset, we identified a Nuclear Factor of Activated T-cells (*McNFAT5*). First, BLASTP searches were performed using the human *NFAT5* protein sequence (UniProt ID: Q9ET54) as the query against the assembled transcriptome to identify putative homologous sequences. Conserved domains specific to *NFAT5*, such as the Rel-like DNA-binding domain and the transcriptional activation domain, were further confirmed using the NCBI Conserved Domain Database (CDD) and SMART analysis. The open reading frame (ORF) of *McNFAT5* was confirmed by sequencing PCR products amplified with two specific primer pairs. The corresponding amino acid sequence was translated, and its theoretical isoelectric point and molecular weight were estimated via ExPASY, while conserved domains were identified using SMART [16]. For phylogenetic analysis, *NFAT5* sequences from invertebrates and vertebrates were obtained from the NCBI gene database. Multiple sequence alignment (MSA) was performed using MAFFT (v7.505), which is a widely recognized tool for the accurate alignment of sequences with low complexity regions. Regions with poor alignment quality or low phylogenetic informativeness were filtered using GBLOCKS to remove background noise. The filtered alignment was then used to construct a phylogenetic tree using the maximum likelihood (ML) method implemented in IQ-TREE, which employs explicit models of molecular evolution. Bootstrap analysis with 5000 replicates was conducted to assess the reliability of each node, and the final tree was presented as a phylogram with branch lengths reflecting evolutionary distances [17].

### 2.3. Bacterial Challenge Experiment, RNA Extraction, and cDNA Synthesis

The bacterial challenge experiment was conducted following the methods described in our previous research [18], In brief, 60 healthy mussels were randomly divided into three treatment groups, each with three replicates, with 20 mussels in each replicate. The adductor muscles were injected with *Vibrio alginolyticus* at a final concentration of 1.0 × 10^7^ cfu/mL with an injection volume of 100 μL. The control group was injected with an equal volume of PBS. *McNFAT5* stimulation times were set at 0 h, 6 h, 12 h, 24 h and 48 h. Total RNA samples were extracted from the hemolymph using RNAiso reagent (TaKaRa, Beijing, China). Subsequently, cDNA was synthesized using the HiScript^®^ II Q Select RT SuperMix for qPCR (+gDNA wiper) kit (Vazyme, Nanjing, China) according to the manufacturer’s instructions.

### 2.4. Quantitative Real-Time PCR Analysis

Quantitative real-time PCR (qRT-PCR) was performed using a 7500 Real-Time PCR System (Applied Biosystems, Waltham, MA, USA) to quantify *McNFAT5* mRNA expression. The primer sequences for RT-qPCR were designed using Primer 5.0 and synthesized by Sangon Bioengineering (Shanghai, China) Co., Ltd., as detailed in Table 1, following previously reported methods. The PCR reaction was conducted in a 20 μL system, which included 0.48 μL of each primer, 10 μL of 2× TB Green Premix Ex Taq II, 2 μL of cDNA template, 0.4 μL of ROX II, and 6 μL of ddH_2_O. The thermocycling program consisted of an initial denaturation at 95 °C for 30 s, which was followed by 40 cycles of 95 °C for 5 s and 60 °C for 34 s. Gene expression levels were analyzed using the 2^−ΔΔCt^ method with β-actin serving as the internal control [19].

### 2.5. Immunohistochemistry

Hemolymph was extracted from three mussels 24 h post-infection with *V. alginolyticus* and preserved in 10% formaldehyde overnight. The samples were dehydrated through a graded ethanol series, and 4 μm tissue sections were prepared using a microtome and mounted on coated slides for immunohistochemical analysis. After deparaffinization and rehydration, the slides were incubated overnight at 37 °C with anti-rWLP antibodies (1:200) in 1% BSA. Detection was performed using a peroxidase-conjugated secondary antibody for rabbit IgG, which was followed by DAB staining. Finally, the sections were examined and imaged using a DFC450C microscope (Leica, Wetzlar, Germany).

### 2.6. RNA Interference

The RNA interference experiment involved designing and synthesizing specific small interfering RNAs (siRNAs) targeting *McNFAT5* along with a control siRNA (siNC) obtained from ShengGong (Shanghai, China) (Table 1). To ensure specificity and minimize off-target effects, the BLAST (https://blast.ncbi.nlm.nih.gov/Blast.cgi accessed on 27 February 2025) online tool was used to screen the designed siRNA sequences for potential non-specific binding with unintended genes. The siRNAs were dissolved in nuclease-free water to a final concentration of 20 mM. For the experiment, 20 mussels infected for 24 h were randomly assigned to two groups: the siNC group and the siRNA group. Each mussel received an injection of 100 μL siRNA solution into the adductor muscle. After 24 h, hemolymph samples were collected from three mussels in each group to evaluate interference efficiency at the mRNA level, using β-actin as an internal control. All samples were tested in triplicate for accuracy and reproducibility.

### 2.7. Measurement of Antioxidant Capacity

To evaluate the impact of exposure to *V. alginolyticus* on the antioxidant capacity of thick-shelled mussels, key biomarkers such as superoxide dismutase (SOD), Na^+^/K^+^-ATPase activity, and total antioxidant capacity (T-AOC) were analyzed [20]. Three mussels were randomly selected from both the control group and each Vibrio-infected group. Hemolymph samples were dissected on ice and collected for subsequent analysis. SOD, Na^+^/K^+^-ATPase, and T-AOC activities were measured using commercial detection kits (A001-3, A070-2-2, and A015-2-1) supplied by the Jiancheng Bioengineering Institute (Nanjing, China) following the manufacturer’s instructions. To ensure the reliability of the results, samples that could not be immediately processed were rapidly frozen in liquid nitrogen and stored at −80 °C with all analyses completed within two weeks. For statistical analysis, a two-tailed *t*-test was employed to assess significant differences between treatment groups and the control group, providing a robust evaluation of treatment effects.

### 2.8. Statistical Analysis

Prior to conducting ANOVA, we tested the assumptions of normality and homogeneity of variance. Normality was assessed using the Shapiro–Wilk test, and homogeneity of variance was evaluated using Levene’s test. All data met these assumptions (*p* > 0.05), allowing parametric analysis. One-way ANOVA was used to analyze the differences in *McNFAT5* expression levels across different tissues following pathogen stimulation. Two-way ANOVA was conducted to evaluate the effects of *V. alginolyticus* infection and *McNFAT5* knockdown on antioxidant enzyme activities, including superoxide dismutase (SOD), Na^+^/K^+^-ATPase, and total antioxidant capacity (T-AOC). Duncan’s multiple comparison test was performed for post hoc analysis. Results are presented as mean ± standard error (SE). The significance level was set at 0.05, and a *p*-value < 0.05 was considered statistically significant.

## 3. Results

### 3.1. McNFAT5 Molecular Characterization

The complete cDNA sequencing of *McNFAT5* revealed that it consists of 1566 amino acids. Using Expasy, *McNFAT5* was found to have a molecular weight of 182.44 kDa, and an isoelectric point of 9.57. SMART (V3.0) analysis predicted RHD and IPT domains in the amino acid sequence (Figure 1A). Additionally, the three-dimensional structure of the *McNFAT5* protein was predicted, displaying the spatial arrangement of its structural elements (Figure 1B). Phylogenetic analysis showed that *McNFAT5* clusters with the *NFAT5* protein of *Mytilus galloprovincialis*, forming a large branch with other mollusks. The clustering of species within the same phylum in the phylogenetic tree indicates that *NFAT5* is highly conserved during evolution (Figure 1C).

### 3.2. Differential McNFAT5 Gene Expression in Different Tissues After Pathogen Stimulation

Agarose gel electrophoresis was used to validate the expression of the *McNFAT5* gene in different tissues of *M. coruscus*. The results showed clear expression bands of *McNFAT5* in all tested tissues, including foot, gill, mantle, hemolymph, gonad and digestive gland, indicating that this gene is widely expressed across various tissues (Figure 2A). After 12 h of exposure to *V. alginolyticus*, the expression and localization of *McNFAT5* protein were assessed using immunofluorescence. DAPI, which stains the nucleus, displayed blue fluorescence. Red fluorescence represented *McNFAT5*. In the control group, weak red fluorescence was observed and was mainly concentrated in the cytoplasm. Following exposure to *V. alginolyticus*, sections of the hemolymph exhibited intense red fluorescence, which was primarily localized in the cell nucleus (Figure 2B). The spatial expression of *McNFAT5* in adult *M. coruscus* after *V. alginolyticus* infection was analyzed via quantitative real-time RT-PCR using the total RNA from six tissues (foot, gill, mantle, hemolymph, gonad, and digestive gland) of 15 individuals. *McNFAT5* was expressed in all tissues at varying levels, with the highest expression in the hemolymph and the digestive gland, moderate levels in the foot, gill, and mantle, and the lowest in the gonad (Figure 2C).

### 3.3. Temporal Expression Patterns of McNFAT5 After V. alginolyticus Stimulation

To investigate the potential biological function of *McNFAT5* in the immune response of *M. coruscus*, the expression levels of *McNFAT5* were measured at various time points following *V. alginolyticus* challenge. The results revealed a significant upregulation of the *McNFAT5* expression at 6 h post-stimulation (*p* < 0.01) with a 3.8-fold increase observed at 12 h (*p* < 0.01). Subsequently, the expression of *McNFAT5* decreased at 24 h and 48 h compared to its peak at 12 h (*p* < 0.01). All data are presented as mean ± SD from three independent biological replicates (Figure 3).

### 3.4. Effects of V. alginolyticus Infection on Antioxidants Activities

In this study, the activities of superoxide dismutase (SOD), Na^+^, K^+^-ATPase, and total antioxidant capacity (T-AOC) in hemolymphs were evaluated following *V. alginolyticus* infection and si*NFAT5* knockdown (Figure 4A). The results demonstrated a significant increase in Na^+^, K^+^-ATPase (Figure 4B), SOD (Figure 4C), and T-AOC (Figure 4D) activities in the infection group. However, the subsequent knockdown of *NFAT5* resulted in a marked reduction in the activity levels of SOD, Na^+^, K^+^-ATPase, and T-AOC.

## 4. Discussion

*NFAT5*, as a critical transcription factor, plays a central role in regulating cellular responses to hypertonic stress and immune activities. In this research, direct RNA sequencing was employed to analyze data from *M. coruscus*, leading to the identification of the transcription factor *NFAT5*. Further analysis using the SMART (version 9.0) software (https://smart.embl.de/, accessed on 27 February 2025) predicted the presence of a highly conserved Rel-homology domain (RHD) and immunoglobulin-like fold domain (IPT) in *McNFAT5*, which is crucial for DNA binding and transcriptional activation. This RHD domain is conserved across a wide range of species from invertebrates to mammals [21]. This domain enables *NFAT5* to specifically bind to DNA target sites and regulate the transcription of genes involved in immune and osmotic stress responses [22]. Similar domains have also been identified in other marine invertebrates, such as oysters [23] and lamprey [24], indicating a conserved function of *NFAT5* across species. The three-dimensional structure further reveals the position of the domains. Phylogenetic analysis further confirmed the evolutionary relationship between *McNFAT5* and other molluscan *NFAT5* proteins. *McNFAT5* clustered closely with other molluscan *NFAT5*s, forming a distinct evolutionary branch within the *NFAT5* family, suggesting a shared ancestral origin and conserved function in mollusks [23]. Together, the results from BLAST (https://blast.ncbi.nlm.nih.gov/Blast.cgi accessed on 27 February 2025) searches, conserved domain analysis, and phylogenetic studies provide strong evidence for the identification and functional characterization of *NFAT5* in *M. coruscus*.

Subcellular localization can determine the specific position of *NFAT5* within the cell [25], such as within the nucleus, cytoplasm, cell membrane, or specific organelles. This localization information is crucial for understanding the function of *NFAT5*. For example, if *NFAT5* is primarily localized within the nucleus, it may be involved in the regulation of gene expression. If localized in the cytoplasm or on the cell membrane, it may participate in processes such as signal transduction or material transport. The immunohistological evaluation of NFATc1 in nearly 300 cases of lymphoma indicates that the majority of tumor lymphocytes express NFATc1 as a cytosolic component despite its absence in classical Hodgkin’s disease and plasma cell proliferations. Particularly intriguing is the discovery of NFATc1 relocating to the nucleus in a minority of lymphoid tumors, potentially reflecting the activation of the NFAT pathway [26]. Under specific physiological or pathological conditions, the subcellular localization of *NFAT5* (Nuclear Factor of Activated T cells 5) shifts from the cytoplasm to the nucleus. For instance, in studies on sepsis-induced acute kidney injury, *NFAT5* has been found to play a crucial role in renal collecting duct cells. When these cells are stimulated by LPS (lipopolysaccharide), Western blot analysis fails to detect an increase in *NFAT5* protein expression levels within the nucleus after LPS stimulation. However, chromatin immunoprecipitation (ChIP) assays reveal that *NFAT5* can bind to the promoter regions of inflammatory cytokines *TNFα* and *MC*P-1, initiating their transcription and protein synthesis. This finding suggests that under LPS stimulation, *NFAT5* may translocate from the cytoplasm to the nucleus to regulate the transcription and synthesis of inflammatory cytokines. We conducted immunofluorescence assays based on studies in vertebrates, and the results showed that the localization changes of *NFAT5* following bacterial infection can be observed: in uninfected cells, *NFAT5* is primarily located in the cytoplasm, while after infection, *NFAT5* signals gradually shift to the nucleus. This indicates that *NFAT5* has been activated and is involved in the immune response.

To clarify the role of *McNFAT5* in regulating innate immune responses, we analyzed its expression patterns post-Vibrio infection using qPCR. Following exposure to *V. alginolyticus*, *McNFAT5* mRNA levels in the hemolymph showed a significant increase, indicating its essential function in initiating and modulating the immune defense against bacterial invasion. This finding is consistent with prior studies. In *B. belcheri*, an NFAT-like gene was identified and cloned, and its evolutionary relationship across different species was analyzed. The transcriptional activity of *NFAT5* was assessed, and through bacterial infection experiments, the impact of *NFAT5* expression on the immune response in *B. belcheri* was observed. The results indicated that the expression of *NFAT5* is correlated with the intensity of the immune response [24]. In invertebrates, *NFAT5* is mainly involved in the cellular response to environmental stress, such as osmotic pressure changes and pathogen invasion. Its function is not limited to cellular stress responses but also includes the regulation of immune defense mechanisms. For example, in insects, *NFAT5* enhances the organism’s resistance to pathogens by regulating the expression of immune-related genes [27]. In contrast to invertebrates, *NFAT5* plays a more complex and diverse role in vertebrates. Studies have shown that *NFAT5* plays an important role in the immune system of vertebrates, particularly in the activation of immune cells and the regulation of immune tolerance. For example, in rheumatoid arthritis (RA) animal models, *NFAT5* has been shown to regulate macrophage function and participate in the disease’s pathogenesis. Specifically, after the activation of macrophages via Toll-like receptors (TLRs), *NFAT5* promotes the production of reactive oxygen species (ROS) and further activates the p38MAPK cascade. Activation of this pathway induces *NFAT5* to secrete CCL2, helping macrophages resist apoptosis, thereby exacerbating the immune response in RA [28]. Upon the stimulation of macrophages by Toll-like receptors (TLRs), *NFAT5* is activated through the xanthine oxidase–reactive oxygen species (ROS)–p38MAPK cascade. This protein provides apoptotic resistance to RA macrophages by inducing CCL2 secretion. Compared to normal tissues, the expression of *NFAT5* is significantly elevated in hepatocellular carcinoma and lung adenocarcinoma cells [29]. The function of *NFAT5* differs between invertebrates and vertebrates. Although *NFAT5* is involved in regulating immune responses in both, in invertebrates, *NFAT5* is more related to cellular responses to osmotic stress and pathogen invasion. In contrast, in vertebrates, *NFAT5* plays a role in more complex immune regulation, including cell death, immune tolerance, and its involvement in chronic inflammation and cancer.

During the infection process of *M. coruscus* by *V. alginolyticus*, hemocytes generate a substantial amount of reactive oxygen species (ROS) to eliminate invading pathogens [30]. At this stage, the expression and activity of superoxide dismutase (SOD) are significantly elevated, mitigating excessive ROS accumulation and preventing oxidative damage to the host [31]. Hemocytes may face challenges such as osmotic imbalance and oxidative stress. The activation of *NFAT5* can regulate the gene expression of Na^+^, K^+^-ATPase, aiding hemocytes in maintaining ionic homeostasis and thereby supporting their phagocytic activity and the secretion of immune factors [32]. Firstly, *NFAT5* influences the activity of Na^+^/K^+^-ATPase by regulating the expression of genes related to ion homeostasis. Several studies have shown that *NFAT5*, by modulating the transcription of specific genes, directly or indirectly affects the function of ion pumps. These genes may be related to the subunits, regulatory factors, or accessory proteins of Na^+^/K^+^-ATPase, thereby influencing its activity. Furthermore, the regulatory role of *NFAT5* may indirectly affect the activity of Na^+^/K^+^-ATPase through oxidative stress response pathways. Specifically, when cells encounter oxidative stress, *NFAT5* helps maintain cellular redox balance by regulating the expression of antioxidant genes, thereby indirectly supporting the normal function of ion pumps. The increased total antioxidant capacity (T-AOC) indicates that *M. coruscus* has initiated a comprehensive antioxidant defense mechanism to counteract the oxidative stress induced by infection. This study demonstrates that *V. alginolyticus* infection causes significant damage to the hemolymph tissue of *M. coruscus*, as evidenced by the increased activities of superoxide dismutase (SOD) and Na^+^, K^+^-ATPase, and total antioxidant capacity (T-AOC) in the hemolymph. This enhancement in antioxidant enzyme activity indicates that the mussels have adopted an adaptive response to cope with the oxidative stress induced by bacterial infection. SOD plays a crucial role in detoxifying superoxide radicals, thereby preventing oxidative damage to cellular components [33]. The elevated levels of T-AOC further reflect the overall capacity of the hemolymph to neutralize reactive oxygen species (ROS) and maintain cellular homeostasis. Interestingly, subsequent knockdown of *NFAT5* resulted in a significant decrease in the activities of SOD, Na^+^, K^+^-ATPase, and T-AOC. This finding suggests that *NFAT5* is a key regulatory factor in the antioxidant defense mechanism of mussels. The transcription factor *NFAT5* may promote the expression of genes encoding antioxidant enzymes and other protective proteins, thereby enhancing the organism’s ability to respond to oxidative stress.

## 5. Conclusions

This study explores *McNFAT5*’s role in immune regulation in *M. coruscus*, especially after *V. alginolyticus* infection. Through a series of experiments, we confirmed that *McNFAT5*, as an essential transcription factor, is significantly upregulated post-infection, leading to the enhanced expression of antioxidant enzyme activities and the strengthening of immune defense mechanisms in the hemolymph. This process aids the host in coping with oxidative stress and pathogen invasion. These findings provide valuable insights into the molecular pathways underlying immune defense in marine invertebrates, emphasizing the fundamental role of *NFAT5* in immune response regulation. Moreover, the results of this study offer a crucial theoretical basis for improving disease resistance in aquaculture species and enhancing their resilience to environmental stressors and pathogens. Future research can further explore the functional roles and regulatory networks of *NFAT5*, enabling the optimization of disease resistance in bivalves and promoting the sustainable development of the aquaculture industry in the face of challenges posed by climate change and the spread of infectious diseases.

## Figures and Tables

**Figure 1 animals-15-00726-f001:**
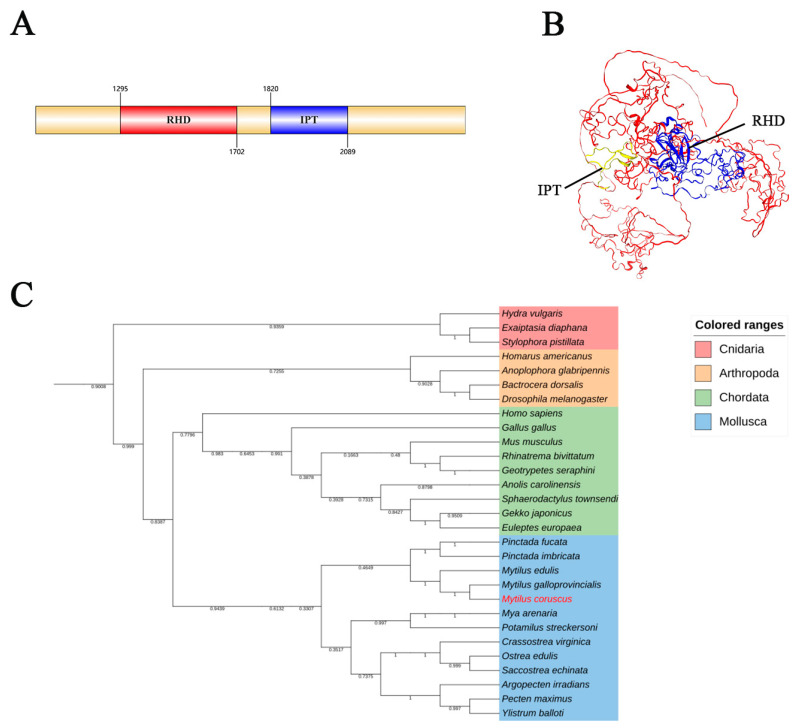
Molecular characterization of *McNFAT5* (**A**) SMART-predicted schematic of *McNFAT5* functional domains. (**B**) SWISS-MODEL-generated three-dimensional structure of *McNFAT5*. (**C**) Phylogenetic analysis of *McNFAT5* with selected species using the neighbor-joining method in MEGA 7.0 with 5000 bootstrap replications. Different colors indicate distinct taxa. The species in red font, *Mytilus coruscus*, represents the focus of our study.

**Figure 2 animals-15-00726-f002:**
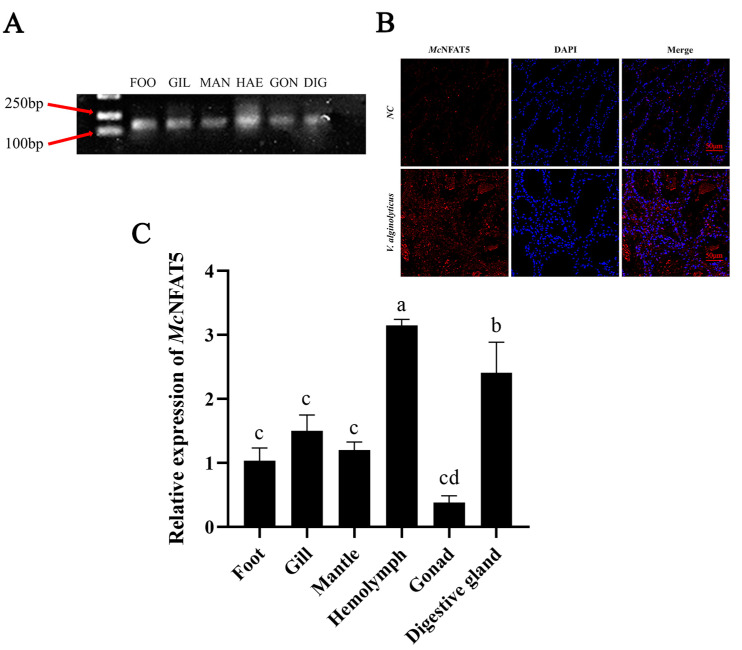
*McNFAT5* responds to pathogen stimulation. (**A**) Agarose gel electrophoresis validation of *McNFAT5* expression in various tissues of *M. coruscus*, including foot, gill, mantle, hemolymph, gonad and digestive gland. The presence of clear bands in all tested tissues confirms the expression of *McNFAT5* across different tissue types. (**B**) Immunofluorescence showing translocation of *NFAT5* to the nucleus after *V. alginolyticus* infection. Red: *McNFAT5* protein, Blue: DAPI-stained nuclei. Scale bars = 20. Each sample was analyzed in triplicate. Vertical error bars represent the mean ± standard error (SE, n = 3). (**C**) The β-actin gene of *M. coruscus* was used as an internal reference to normalize the cDNA templates across all samples. Differential expression across the six tissues was analyzed using one-way ANOVA. Distinct letters indicate statistically significant differences (*p* < 0.05) in the relative expression levels of *McNFAT5* mRNA.

**Figure 3 animals-15-00726-f003:**
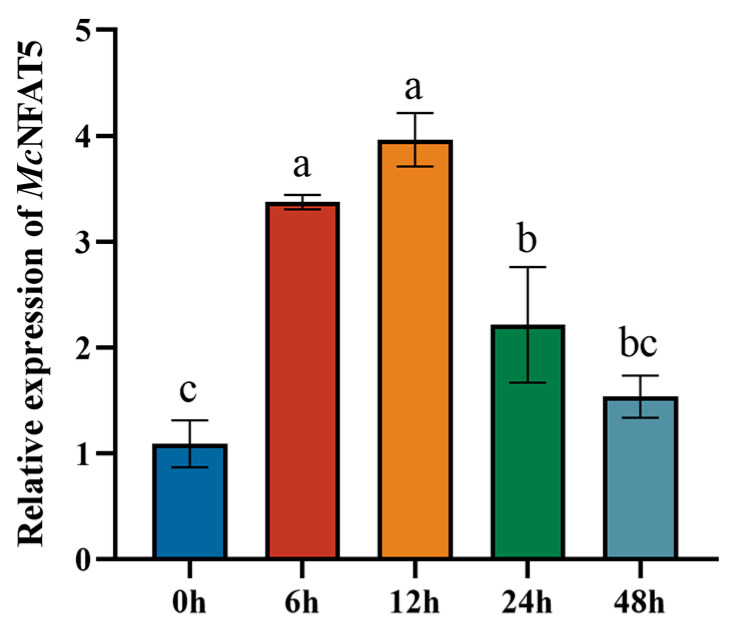
Temporal expression patterns of *McNFAT5* after infection. *McNFAT5* mRNA expression was assessed via qRT-PCR at 6 h, 12 h, 24 h, and 48 h post-*V. alginolyticus* infection. Data are shown as mean ± SD (n = 3) with different letters indicating significant differences (*p* < 0.05).

**Figure 4 animals-15-00726-f004:**
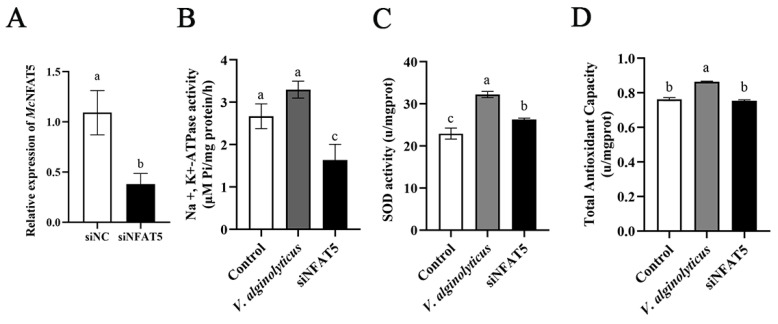
The impact of *NFAT5* knockdown on the activity of three key enzymes and antioxidant capacity. (**A**) The expression of *McNFAT5* was downregulated by Si-*McNFAT5*. (**B**) Na^+^, K^+^-ATPase. (**C**) SOD. (**D**) T-AOC. The vertical bars represent the mean ± SD (n = 3). Different letters indicate significant differences (*p* < 0.05).

**Table 1 animals-15-00726-t001:** PCR primer pairs used in the present study.

Primer	Sequences (5′–3′)	Usage
*McNFAT5*-F	TCTTCTTGACCGTGCTGGAC	qRT–PCR
*McNFAT5*-R	TCGTCGGACTTTTGGCACTT	
McTRAF-F	TGTGCCAATTCCCTGTCCT	qRT–PCR
McTRAF-R	GGACACTCTTTATGCAGG	
McIRAK-F	CCTTTTATGGCAGCAGCGTG	qRT–PCR
McIRAK-R	AAAATCCAGTGCCCGATGGT	
Mcmyticofensin-F	TGTGGCTCTAGAAGTTGCTGATG	qRT–PCR
Mc myticofensin-R	TCAATCTGAACCAGCCTCCAC	
β-actin-F	GCTACGAATTACCTGACGGAC	qRT–PCR
β-actin-R	TTCCCAAGAAAGATGGTTGTAACAT	
siNC	UUCUCCGAACGUGUCACGUTT	RNAi
	ACGUGACACGUUCGGAGAATT	
Si*NFAT5*	GACAAUAAAUCAACUGUUATT	RNAi
	UAACAGUUGAUUUAUUGUCTT	

## Data Availability

No datasets were generated or analyzed during the current study.

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
