# Peer review of "The Role of NFAT5 in Immune Response and Antioxidant Defense in the Thick-Shelled Mussel (Mytilus coruscus)"

_animals, 2025, doi:10.3390/ani15050726_

Round 1

Reviewer 1 Report

Comments and Suggestions for Authors

This manuscript presents a novel investigation into the role of the NFAT5 gene in the immune response and antioxidant defense of the thick-shelled mussel (Mytilus coruscus). The discovery that NFAT5 plays a crucial role in defense against Vibrio alginolyticus infections provides new insights into the mechanisms of immune regulation in marine bivalves. This highlights the importance of the study.

Overall, the manuscript is suitable for publication in Animals. The introduction is well written and addresses all aspects necessary for manuscript compression. The material and methods are adequate, the results and discussion are appropriate, as well as the conclusion.

Main comments.

Line 210: The statistical analysis section describes the general approach well, but could be improved to include specific details on the normality and homogeneity tests performed. Which tests were used?

My recommendation: minor revisions.

Author Response

comments: Line 210: The statistical analysis section describes the general approach well, but could be improved to include specific details on the normality and homogeneity tests performed. Which tests were used?

Reply: Thank you. We have provided a more detailed description of the statistical analysis in the revised manuscript. Prior to conducting ANOVA, we tested the assumptions to ensure the validity of the statistical analysis: Shapiro-Wilk test was used to assess the normality of the data. Levene’s test was used to evaluate the homogeneity of variance. The results indicated that all data met the assumptions of normality and homogeneity of variance (p > 0.05), allowing for parametric analysis. One-way ANOVA was used to analyze the differences in McNFAT5 expression levels across different tissues following pathogen stimulation (Figure 2C).Two-way ANOVA was performed to evaluate the effects of Vibrio alginolyticus infection and McNFAT5 knockdown on antioxidant enzyme activities, including superoxide dismutase (SOD), Na⁺/K⁺-ATPase, and total antioxidant capacity (T-AOC). Duncan’s multiple comparison test was applied for post hoc analysis. We have incorporated these details into the revised manuscript to improve the clarity and rigor of the statistical analysis. The specific modifications are as follows:

Line210-220: Prior to conducting ANOVA, we tested the assumptions of normality and homogeneity of variance. Normality was assessed using the Shapiro-Wilk test, and homogeneity of variance was evaluated using the Levene’s test. All data met these assumptions (p > 0.05), allowing parametric analysis. One-way ANOVA was used to analyze the differences in McNFAT5 expression levels across different tissues following pathogen stimulation. Two-way ANOVA was conducted to evaluate the effects of V. alginolyticus infection and McNFAT5 knockdown on antioxidant enzyme activities, including superoxide dismutase (SOD), Na⁺/K⁺-ATPase, and total antioxidant capacity (T-AOC). Duncan’s multiple comparison test was performed for post hoc analysis. Results are presented as mean ± standard error (SE). The significance level was set at 0.05, and a p-value < 0.05 was considered statistically significant.

Reviewer 2 Report

Comments and Suggestions for Authors

This study provides the first characterization of NFAT5 in Mytilus coruscus, investigating its role in immune regulation and antioxidant defense in response to Vibrio alginolyticus infection. The results demonstrate that McNFAT5 is highly expressed in hemolymph, plays a critical role in immune response, and influences key antioxidant enzymes (SOD, Na⁺/K⁺-ATPase, and T-AOC). However, there are several areas that require clarification and improvement before the manuscript can be considered for publication.

  1. The immunofluorescence images (Figure 2B) need better contrast and resolution to enhance the visualization of NFAT5 nuclear translocation.
  2. Ensure that sample sizes (n values) are consistent between figure legends and the main text.
  3. While siRNA knockdown of McNFAT5 was performed, it would strengthen the findings if a McNFAT5 overexpression group was included to further confirm its function.
  4. The role of NFAT5 in immune signaling pathways should be further elaborated, especially in comparison with other invertebrate and vertebrate species.
  5. How does NFAT5 regulate Na⁺/K⁺-ATPase activity at the molecular level? Does it directly influence ion transport genes or oxidative stress response pathways?
  6. Ensure gene and protein names follow a consistent format (e.g., italics for gene names, normal text for proteins).
Comments on the Quality of English Language

Some sentences are lengthy and could be streamlined for better readability.

Author Response

Comments:  This study provides the first characterization of NFAT5 in Mytilus coruscus, investigating its role in immune regulation and antioxidant defense in response to Vibrio alginolyticus infection. The results demonstrate that McNFAT5 is highly expressed in hemolymph, plays a critical role in immune response, and influences key antioxidant enzymes (SOD, Na⁺/K⁺-ATPase, and T-AOC). However, there are several areas that require clarification and improvement before the manuscript can be considered for publication.

1.Comments: The immunofluorescence images (Figure 2B) need better contrast and resolution to enhance the visualization of NFAT5 nuclear translocation.

Reply: Thank you. We have enhanced the contrast and resolution of Figure 2B to improve the visualization of NFAT5 nuclear localization. Specifically, we employed advanced image processing techniques using Adobe Photoshop and ImageJ, including contrast enhancement and noise reduction, to further refine image clarity. The revised Figure 2B now provides a more precise and detailed depiction of NFAT5 nuclear localization.

2.Comments: Ensure that sample sizes (n values) are consistent between figure legends and the main text.

Reply: Thank you for your valuable feedback. We have carefully checked the consistency of sample sizes (n values) between the figure legends and the main text. After thorough verification, we have made the necessary adjustments to ensure that all reported sample sizes are consistent across the manuscript.

3.Comments: While siRNA knockdown of McNFAT5 was performed, it would strengthen the findings if a McNFAT5 overexpression group was included to further confirm its function.

Reply: Thank you. This study primarily focused on functional loss analysis through RNA interference (RNAi) to investigate the role of McNFAT5 in immune regulation. Currently, establishing an efficient gene overexpression system in Mytilus coruscus remains technically challenging due to the lack of a well-developed genetic manipulation platform for this species. As a result, an overexpression experiment was not included within the scope of this study.

We also recognize the importance of functional validation through both gene knockdown and overexpression approaches. Therefore, in future studies, we plan to develop an overexpression system suitable for Mytilus coruscus to further verify the function of McNFAT5 and explore its specific regulatory mechanisms in greater depth.

We sincerely appreciate your valuable feedback, which has provided important guidance for refining our research approach and shaping the direction of our future studies.

4. Comments: The role of NFAT5 in immune signaling pathways should be further elaborated, especially in comparison with other invertebrate and vertebrate species.

Reply: Thank you. We have provided a more detailed discussion of the function of NFAT5 in the revised version, with a particular focus on comparing the role of NFAT5 in immune responses across different invertebrate and vertebrate species. The specific revisions are as follows:

Line345-358: In invertebrates, NFAT5 is mainly involved in the cellular response to environmental stress, such as osmotic pressure changes and pathogen invasion. Its function is not limited to cellular stress responses but also includes the regulation of immune defense mechanisms. For example, in insects, NFAT5 enhances the organism's resistance to pathogens by regulating the expression of immune-related genes. In contrast to invertebrates, NFAT5 plays a more complex and diverse role in vertebrates. Studies have shown that NFAT5 plays an important role in the immune system of vertebrates, particularly in the activation of immune cells and the regulation of immune tolerance. For example, in rheumatoid arthritis (RA) animal models, NFAT5 has been shown to regulate macrophage function and participate in the disease's pathogenesis. Specifically, after activation of macrophages via Toll-like receptors (TLRs), NFAT5 promotes the production of reactive oxygen species (ROS) and further activates the p38MAPK cascade. Activation of this pathway induces NFAT5 to secrete CCL2, helping macrophages resist apoptosis, thereby exacerbating the immune response in RA.

Line363-368: The function of NFAT5 differs between invertebrates and vertebrates. Although NFAT5 is involved in regulating immune responses in both, in invertebrates, NFAT5 is more related to cellular responses to osmotic stress and pathogen invasion. In contrast, in vertebrates, NFAT5 plays a role in more complex immune regulation, including cell death, immune tolerance, and its involvement in chronic inflammation and cancer.

5. Comments: How does NFAT5 regulate Na⁺/K⁺-ATPase activity at the molecular level? Does it directly influence ion transport genes or oxidative stress response pathways?

Reply: Thank you. NFAT5 regulates Na⁺/K⁺-ATPase primarily by modulating the expression of genes related to ion homeostasis and oxidative stress response, rather than directly acting on the ion pump itself. At the molecular level, studies have shown that NFAT5 can regulate the transcription of specific genes, which may encode Na⁺/K⁺-ATPase subunits, regulatory factors, or accessory proteins, thereby indirectly influencing its activity. This suggests that NFAT5 plays a crucial role in maintaining ion balance by modulating the expression of ion transport-related genes. Additionally, NFAT5 plays a key role in the oxidative stress response pathway, further affecting Na⁺/K⁺-ATPase activity. The specific modifications are as follows:

Line376-385: Firstly, NFAT5 influences the activity of Na⁺/K⁺-ATPase by regulating the expression of genes related to ion homeostasis. Several studies have shown that NFAT5, by modulating the transcription of specific genes, directly or indirectly affects the function of ion pumps. These genes may be related to the subunits, regulatory factors, or accessory proteins of Na⁺/K⁺-ATPase, thereby influencing its activity. Furthermore, the regulatory role of NFAT5 may indirectly affect the activity of Na⁺/K⁺-ATPase through oxidative stress response pathways. Specifically, when cells encounter oxidative stress, NFAT5 helps maintain cellular redox balance by regulating the expression of antioxidant genes, thereby indirectly supporting the normal function of ion pumps.

6. Comments: Ensure gene and protein names follow a consistent format (e.g., italics for gene names, normal text for proteins).

Reply: Thank you.We have thoroughly checked the manuscript and ensured that: Gene names are consistently italicized; Protein names are written in normal text.

Reviewer 3 Report

Comments and Suggestions for Authors

The topic of the manuscript is novel and interesting especially for aquatic toxicologists. 

I consider the manuscript's strengths to be a very well-written introduction and a concise, solid discussion. The Abstract and Simple Summary are legible, clear, and easy to understand.

The following require supplementation and explanation:

1) experimental conditions,

2) number of animals used for analysis,

3) statistical methods.

The authors did not provide basic water parameters, i.e. concentration of nitrates, nitrites, ammonia, phosphates and pH. Therefore, it should be explained why these parameters were not measured. In addition, the authors should admit in the discussion that any exceedances could have had an adverse effect on the obtained results. The authors may also recommend including these parameters in next experiments (regardless of whether it will be conducted by the authors or other researchers).
It should be clarified whether the given salinity is NaCl or all salts combined (simultaneously).

Please specify exactly how many individuals were used for a given analysis. Isn't n=3 for immunohistochemistry too low value? Please justify.

The description of the statistical analysis is imprecise. We do not know whether the authors tested the assumptions of ANOVA (normality of distribution, homogeneity of variance) and what were the results of such an analysis (if it was performed). We also do not know which data were analyzed using one-way ANOVA and which using two-way ANOVA.

The authors incorrectly state the significance level. It should be written: "The significance level was 0.05". The significance level is not the same as p-value and should be precisely defined, and not written that it was lower than something. The significance level is equal to 0.05 in this type of research.

Apart from the above comments, I have no further comments regarding this manuscript.

Author Response

Comments: The topic of the manuscript is novel and interesting especially for aquatic toxicologists.

I consider the manuscript's strengths to be a very well-written introduction and a concise, solid discussion. The Abstract and Simple Summary are legible, clear, and easy to understand.

The following require supplementation and explanation:

1) experimental conditions,

2) number of animals used for analysis,

3) statistical methods.

The authors did not provide basic water parameters, i.e. concentration of nitrates, nitrites, ammonia, phosphates and pH. Therefore, it should be explained why these parameters were not measured. In addition, the authors should admit in the discussion that any exceedances could have had an adverse effect on the obtained results. The authors may also recommend including these parameters in next experiments (regardless of whether it will be conducted by the authors or other researchers).
It should be clarified whether the given salinity is NaCl or all salts combined (simultaneously).

Reply : Thank you. In this study, we cultured mussels under controlled laboratory conditions and managed regular water changes and filtration systems to ensure the stability of the water environment. We monitored key environmental parameters such as water temperature and salinity but did not measure specific water quality indicators such as nitrate, nitrite, ammonia, and phosphate. The primary reason for this omission is that our study focuses on immune responses rather than the effects of environmental stressors. However, we acknowledge that these parameters may have a potential impact on the physiological state of the mussels. We have added a statement in the Animals section recognizing that if these water quality parameters exceed the standard levels, they could influence the experimental results. Future studies should incorporate these water quality indicators to comprehensively assess the effects of environmental factors on the mussel immune system.

The salinity values provided in this study refer to the total dissolved salts, not just NaCl. We have clarified this in the revised manuscript. The specific modifications are as follows:

Line124: Add “(representing the total dissolved salts, not just NaCl)”

Comments: Please specify exactly how many individuals were used for a given analysis. Isn't n=3 for immunohistochemistry too low value? Please justify.

Reply : Thank you. We have explicitly stated the number of individuals (n) used for each analysis in the revised manuscript. In the immunohistochemistry experiment, n = 3 was used to detect McNFAT5 protein localization after V. alginolyticus infection. Immunohistochemistry is primarily a qualitative method used to examine the spatial localization of proteins rather than for quantitative analysis. A sample size of n = 3 is commonly used in invertebrate studies, and several related studies have employed the same approach for protein localization validation. Additionally, multiple sections were examined per individual to ensure the stability of the experimental results and to minimize individual variability. The specific modifications are as follows:

Comments: Line154: Replace: “Mussels infected for 24 hours were divided into two groups”with “20 mussels infected for 24 hours were divided into two groups”

The description of the statistical analysis is imprecise. We do not know whether the authors tested the assumptions of ANOVA (normality of distribution, homogeneity of variance) and what were the results of such an analysis (if it was performed). We also do not know which data were analyzed using one-way ANOVA and which using two-way ANOVA.

Relpy: Thank you for your valuable feedback. We have provided a more detailed description of the statistical analysis in the revised manuscript. Prior to conducting ANOVA, we tested the assumptions to ensure the validity of the statistical analysis: Shapiro-Wilk test was used to assess the normality of the data. Levene’s test was used to evaluate the homogeneity of variance. The results indicated that all data met the assumptions of normality and homogeneity of variance (p > 0.05), allowing for parametric analysis. One-way ANOVA was used to analyze the differences in McNFAT5 expression levels across different tissues following pathogen stimulation (Figure 2C).Two-way ANOVA was performed to evaluate the effects of Vibrio alginolyticus infection and McNFAT5 knockdown on antioxidant enzyme activities, including superoxide dismutase (SOD), Na⁺/K⁺-ATPase, and total antioxidant capacity (T-AOC). Duncan’s multiple comparison test was applied for post hoc analysis. We have incorporated these details into the revised manuscript to improve the clarity and rigor of the statistical analysis. The specific modifications are as follows:

Line210-220: Prior to conducting ANOVA, we tested the assumptions of normality and homogeneity of variance. Normality was assessed using the Shapiro-Wilk test, and homogeneity of variance was evaluated using the Levene’s test. All data met these assumptions (p > 0.05), allowing parametric analysis. One-way ANOVA was used to analyze the differences in McNFAT5 expression levels across different tissues following pathogen stimulation. Two-way ANOVA was conducted to evaluate the effects of V. alginolyticus infection and McNFAT5 knockdown on antioxidant enzyme activities, including superoxide dismutase (SOD), Na⁺/K⁺-ATPase, and total antioxidant capacity (T-AOC). Duncan’s multiple comparison test was performed for post hoc analysis. Results are presented as mean ± standard error (SE). The significance level was set at 0.05, and a p-value < 0.05 was considered statistically significant.

Comments: The authors incorrectly state the significance level. It should be written: "The significance level was 0.05". The significance level is not the same as p-value and should be precisely defined, and not written that it was lower than something. The significance level is equal to 0.05 in this type of research.

Reply: Thank you. We have acknowledged the distinction between the significance level and the p-value and have made the necessary revisions in the revised manuscript. The specific modifications are as follows:

Line219-220: The significance level was set at 0.05, and a p-value < 0.05 was considered statistically significant.

Apart from the above comments, I have no further comments regarding this manuscript.
